# Energy Saving Strategy of UAV in MEC Based on Deep Reinforcement Learning

**Zhiqiang Dai \*, Gaochao Xu, Ziqi Liu, Jiaqi Ge \* and Wei Wang**

Department of Computer Science and Technology, Jilin University, Changchun 130012, China; xugc@jlu.edu.cn (G.X.); ziqi20@mails.jlu.edu.cn (Z.L.); wangw19@mails.jlu.edu.cn (W.W.)
\* Correspondence: daizq19@mails.jlu.edu.cn (Z.D.); gejq18@mails.jlu.edu.cn (J.G.)

**Abstract:** Unmanned aerial vehicles (UAVs) have the characteristics of portability, safety, and strong adaptability. In the case of a maritime disaster, they can be used for personnel search and rescue, real-time monitoring, and disaster assessment. However, the power, computing power, and other resources of UAVs are often limited. Therefore, this paper combines a UAV and mobile edge computing (MEC), and designs a deep reinforcement learning-based online task offloading (DOTO) algorithm. The algorithm can obtain an online offloading strategy that maximizes the residual energy of the UAV by jointly optimizing the UAV's time and communication resources. The DOTO algorithm adopts time division multiple access (TDMA) to offload and schedule the UAV computing task, integrates wireless power transfer (WPT) to supply power to the UAV, calculates the residual energy corresponding to the offloading action through the convex optimization method, and uses an adaptive *K* method to reduce the computational complexity of the algorithm. The simulation results show that the DOTO algorithm proposed in this paper for the energy-saving goal of maximizing the residual energy of UAVs in MEC can provide the UAV with an online task offloading strategy that is superior to other traditional benchmark schemes. In particular, when an individual UAV exits the system due to insufficient power or failure, or a new UAV is connected to the system, it can perform timely and automatic adjustment without manual participation, and has good stability and adaptability.

**Keywords:** UAV; MEC; TDMA; WPT; convex optimization; deep reinforcement learning



## 1. Introduction

With the development of the economy, there has been an increase in activities such as maritime trade, fishing, and drilling platforms, but marine accidents also occur frequently. Due to the complex sea conditions, the location of ships and personnel in distress is not fixed, and it is difficult to carry out search and rescue work. Moreover, it is difficult to achieve the ideal rescue effect only by traditional methods such as ship cruising and people watching. The Internet of Things (IoT) has been integrated into every aspect of our lives [1,2]. Among them, unmanned aerial vehicles (UAVs) [3] have the advantages of being unmanned and lightweight, with a quick response and strong adaptability. They cooperate with ships and use video image technology during the day and infrared detection technology at night [4–6], which can provide 24 h uninterrupted personnel search and rescue, disaster monitoring, material delivery and other services for rescue work. Efficiency will be greatly improved. Thus, in recent years, many countries have vigorously developed UAVs that can perform maritime search and rescue missions. However, resources such as the power and computing power of UAVs are often limited [7,8], which has become a key problem to be solved in UAV applications. Therefore, this paper combines a UAV with mobile edge computing (MEC) to provide a solution to the problem of UAV battery life.

MEC [9,10] is to set up an edge server with more sufficient computing power and bandwidth near the user equipment to provide computing resources for UAVs and speed up data processing. The computing task offloading of UAVs can adopt partial offloading

or binary offloading methods [11]. Partial offloading [12] means that the computing data can be split into two parts: one part is executed locally and the other part is offloaded to the edge server for execution. Binary offloading [13] is suitable for the case where the computing task cannot be split. The whole task is either executed locally or offloaded to the edge server for execution. On the basis of reducing the energy consumption of a UAV itself, by incorporating radio frequency (RF)-based [14] wireless power transfer (WPT) [15] into MEC, it can continuously provide energy for the UAV. WPT uses an energy transmitter to wirelessly broadcast energy to the UAV [16,17]. The UAV can use the energy for computing or offloading, and the residual energy can also be converted into equipment power, thereby extending the working time of the equipment. In the scenario of multiple UAVs, this is generally achieved through the joint optimization of the UAV computing task offloading decision, offloading time and other resources, which is transformed into a mixed integer programming (MIP) problem for a solution [18–23]. The algorithms used mainly include the coordinate descent method, heuristic algorithm, convex optimization method and convex relaxation technique. However, in some cases, the computational complexity of these algorithms may increase rapidly, which may affect the decision-making effect. Moreover, traditional offline offloading strategies are not suitable for fast fading channels. Therefore, they cannot meet the requirements of ultra-low delay.

Deep reinforcement learning [24,25] is suitable for dealing with variable state space and high-dimensional data, and has a strong fitting effect and learning ability. It can obtain the online offloading strategy according to the actual wireless channel environment changes. The online algorithm is practical because the decision of each time block is made under the condition of a random channel and unknown computing data [26]. Therefore, this paper proposes a deep reinforcement learning-based online task offloading (DOTO) algorithm that can prolong the working time of UAVs. The main contributions of this paper are as follows:

(1) The MEC system that we consider consists of a multi-antenna AP with an edge server placed on the ship and multiple UAVs with a single antenna. The AP can provide wireless energy for the UAV and can also be used to receive UAV computing tasks. The UAV computing task adopts the binary offload method, and the offload scheduling is implemented based on the time division multiple access (TDMA) communication protocol. In order to achieve the research goal of maximizing the residual energy of the UAV, we propose an optimization scheme to jointly optimize the UAV time and communication resources. The research objective is formalized as the problem of maximizing the residual energy of UAV.

(2) The formalized problem of maximizing the residual energy is a non-convex problem and difficult to solve. However, once the offloading decision is given, the problem can be transformed into a convex problem, which can be solved by a convex optimization method. Therefore, we split the target problem into two sub-problems: the time resource allocation problem and task offloading problem. Firstly, we use the convex optimization method to solve the time resource allocation problem, and then obtain an online task offloading strategy that maximizes the residual energy of the UAV in wireless fading environments based on the DOTO algorithm.

(3) To reduce the computational complexity of the DOTO algorithm, we propose a new adaptive $K$ quantization method, which reduces the quantization action of the algorithm with the increase in the time block. Under the condition of ensuring the quality of the offloading strategy, the exponentially increased delay due to the increase in the number of devices is reduced so as to be almost unchanged.

(4) Simulation experiments show that the DOTO algorithm proposed for the research objective of maximizing the UAV's residual energy in MEC can provide users with an online computing offloading strategy that is superior to other traditional benchmark schemes. Moreover, when there is a UAV in the MEC system that exits the system due to insufficient power or failure, or if a new UAV is connected to the system, the

DOTO algorithm can still quickly converge to provide an effective online computing task offloading strategy for the MEC system in time.

The remainder of the article is organized as follows. The second section discusses related work. The third section introduces the system model. The fourth section provides the problem formulation and solution. The fifth section describes the DOTO algorithm. The sixth section provides the simulation results. The seventh section gives the conclusions.

## 2. Related Work

MEC has the characteristics of strong real-time performance, ultra-high bandwidth and ultra-low delay [27]. It provides a solution to the problem of the limited computing and battery life of wireless devices such as UAVs. Thus, it is a very promising technology. By jointly optimizing communication and computing resources, the computing power and battery life of wireless devices can be effectively improved [18]. In the literature [19], scholars have jointly optimized communication resources, computing task offloading, CPU and other resources. The problem is solved by using the distributed convex optimization method and alternating direction multiplier method. In the literature [20], scholars use the coordinate descent method (CD) in a non-gradient optimization algorithm. The CD algorithm will fix the dimensions of other variables when looking for the optimal solution, and only search in the direction of one variable. It is a simple but very efficient algorithm. However, the disadvantage is that if there is a strong correlation between multiple variables, the search will be very slow, and the corresponding search results will be poor. In the literature [21], scholars proposed a heuristic algorithm with low complexity and adopted convex optimization technology for the resource allocation problem. However, in the process of offloading strategy optimization, with the increase in wireless devices, the search space grows exponentially, and the convergence time of the algorithm decreases significantly. In the literature [22], a heuristic search algorithm based on convex relaxation is studied, which makes it continuous from 0 to 1 by relaxing integer variables. In the literature [23], a positive semi-definite relaxation method is employed by adopting quadratic constrained quadratic optimization. However, the method based on convex relaxation needs many iterations to find the qualified local optimal solution. The search time is long, and it is not suitable for fast fading channels requiring ultra-low delay.

In the wireless fading environment, the optimal offloading decision of the wirelessly powered MEC system will be affected [28]. Moreover, the traditional offline offloading decision is no longer applicable in practical scenarios, because it cannot make timely and correct uninstallation decisions. TDMA can divide time into small blocks wherein wireless channels do not interfere with each other, and, combined with MEC, can well overcome the effects of wireless fading [29]. In the literature [30], based on TDMA, scholars solved the problem of minimizing the total energy consumption of MEC by jointly optimizing the energy transmission beam, CPU frequency and computing task offload ratio of edge servers. In the literature [31], a two-stage algorithm and a three-stage alternative algorithm are proposed for solving the computation rate maximization problems in a UAV-enabled MEC wireless-powered system.

At the same time, in order to continuously provide energy for wireless devices, RF-based WPT technology can be integrated into the MEC. WPT technology is simple, safe, highly adaptable to the environment, very convenient to use and widely used. In the literature [8], scholars studied the problem of extending the battery life of wireless devices through energy harvesting and resource allocation in the WPT-MEC system. Deep reinforcement learning can update the offloading strategy in real time according to wireless channel changes [32,33]. An offloading strategy based on deep Q-network (DQN) was proposed in [34] to optimize computation, but DQN is not suitable for the case of too many wireless devices. In the literature [35], a cooperative multi-agent deep reinforcement learning framework is investigated. By jointly designing the trajectories, computation task allocation and communication resource management of UAVs, the sum of execution delay and energy consumption is reduced. In the literature [36], scholars propose a novel deep

reinforcement learning method to optimize UAV trajectory controlling and users' offloaded task ratio scheduling and improve the performance of the UAV-assisted MEC system. It maximizes the system's stability and minimizes the energy consumption and computation latency of the UAV-assisted MEC system.

In previous studies, most researchers focus on the partial offloading mode. For energy-saving goals, most of them focus on the energy consumption of the equipment itself, and there is very little work using deep reinforcement learning methods. In contrast, we increase the energy received by the wireless device on the basis of reducing the energy consumption of the wireless device. We propose a new objective problem of maximizing the residual energy of wireless devices in binary offload mode. Using deep reinforcement learning technology, the optimal MEC online computing offloading strategy is finally obtained.

### 3. System Model

This paper considers a wireless-powered MEC network system, which is composed of a multi-antenna AP integrated with an edge server and a set $N = \{1, \ldots, N\}$ of single-antenna UAVs placed on the ship. The AP has a stable power supply and higher computing power than the UAV. The AP can not only receive data from the UAV for calculation, but also transmit power to the UAV through WPT. The computing tasks of the UAV can be executed locally or offloaded to the MEC for computing, following by the transmission of the returned results. At the same time, the wireless energy transmitted by the harvesting AP can be used for local computing or task offloading, and the residual energy can be used to charge the battery. The system model is shown in Figure 1.

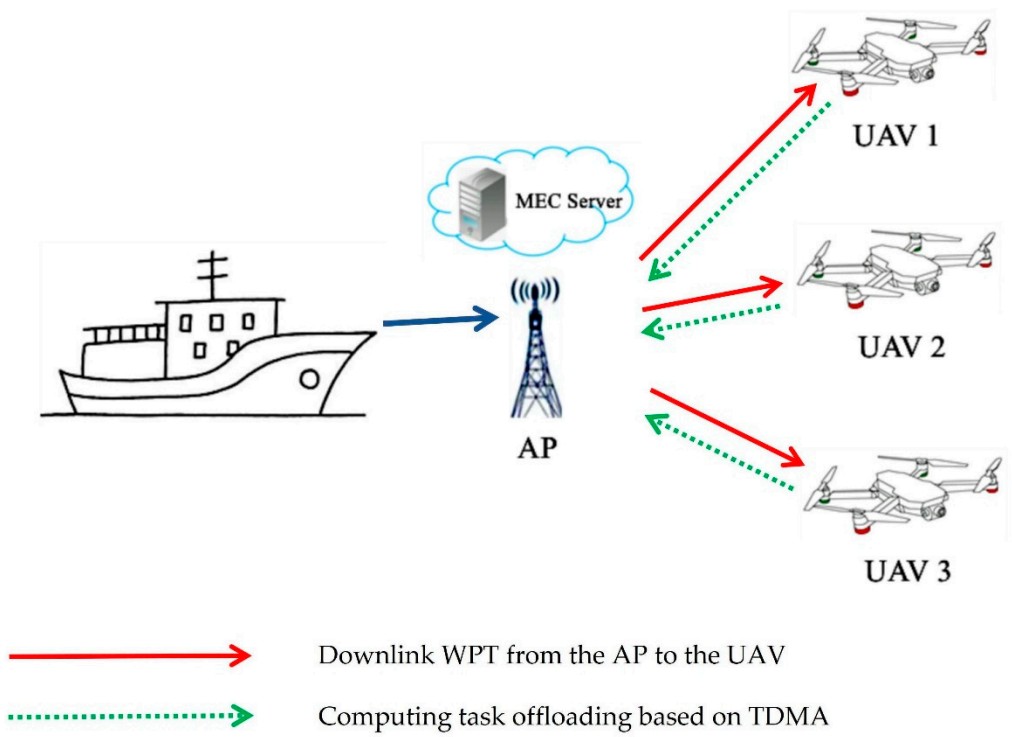

**Figure 1.** System model.

The UAV computing task adopts a binary offloading decision. In other words, all tasks are executed locally, as shown in UAV 1 in Figure 1, or all of them are offloaded to the AP for execution, as shown in Figure 1 for UAV 2 and UAV 3. The indicator variable is represented by $u_i \in \{0, 1\}$, where $u_i = 0$ indicates that the task is computed locally, and $u_i = 1$ indicates that the computing task of the $i-$th UAV is offloaded to the AP for execution. We use $U = \{1, \ldots, N\}$ to represent the set of all UAVs, where $N$ represents a total of $N$ UAVs in the system. $U_0$ denotes the set of UAVs whose computing mode is local

computing, and $U_1$ denotes the set of UAVs whose computing mode is offloaded to the edge server, so it is mutually exclusive with the two sets and has $U_0 \cup U_1 = U$.

### 3.1. Energy Transfer Model

In the system, we use TDMA to offload and schedule computing tasks. The system time distribution is shown in Figure 2. The system time is divided into consecutive time blocks with length $T$, which is set smaller than the channel mutual interference time—for example, only on the scale of seconds [37]. At this time, wireless power transmission and task offloading are performed in the same frequency band. Let $c_n$ denote the wireless channel gain of the AP and the $n-$th UAV in the same time block. The communication speed and energy acquisition between the AP and the UAV are both related to the wireless channel gain. Since the mutual interference between wireless power transmission and communication is avoided, the wireless channel gain remains constant within each time block, but may change between different time blocks [38]. At the beginning of each time block, the duration of AP transmitting energy to UAV is $a\,T$, $a \in [0,1]$. Therefore, the energy captured by the $n-$th UAV is:

$$E_n = \mu\,P\,c_n\,a\,T, n = 1, \ldots, N \tag{1}$$

where $\mu \in (0,1)$ denotes the energy harvesting efficiency and $P$ denotes the AP transmit power. Using the harvested energy, each UAV needs to complete the computing task before the end of a time block, and the calculation of the task is executed immediately from the beginning of the time block. The offloading time of the $j-$th UAV is $b_j\,T$, $b \in [0\,,\,1]$. Here, we assume that the computing speed and transmit power of the AP are much larger than those of resource-constrained UAVs, such as more than three orders of magnitude [28,30], so we can safely ignore the time for AP computing tasks. Since the calculation result data of UAVs are usually much smaller than those for the offloaded calculation task, we can also safely ignore the calculation result download time.

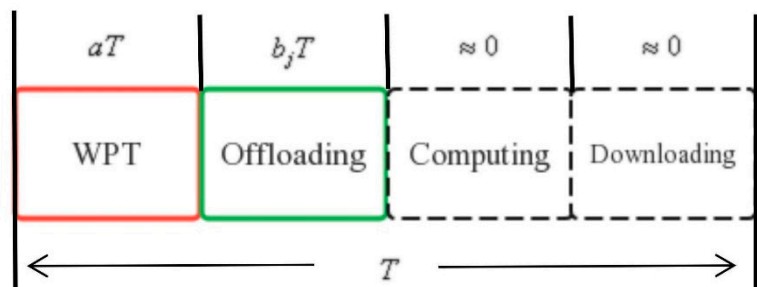

**Figure 2.** System time allocation.

Since each time block is only occupied by WPT and task offloading, there is

$$a + \sum_{j \in U_1} b_j \leq 1 \tag{2}$$

### 3.2. Local Computing Model

The UAV in the local computing mode can harvest energy from the AP and compute its tasks simultaneously. Thus, the UAV can continue computing for the entire time block $T$. Let $O$ denote the number of cycles required by the processor to process one bit of task data, and its size depends on the nature of the computing task. Let $R_i$ denote the computing task size (bit) of the $i-$th UAV, and the number of computing bits $R_i^{loc}$ and $R_i$ in the local computing mode are equal. Moreover, the relationship between $R_i^{loc}$ and CPU computing speed $f_i$ is:

$$R_i^{loc} = R_i = \frac{f_i\,t_i}{O}, \; i \in U_0 \tag{3}$$

Let $k_i$ denote the calculated energy efficiency coefficient of the processor, which depends on the chip architecture of the UAV [18]. Therefore, the energy consumption model of the $i-$th UAV is:

$$E_i^{loc} = k_i f_i^3 t_i, \ i \in U_0 \tag{4}$$

In order to ensure the sustainable operation of the UAV, the energy consumed by the UAV shall be constrained by the energy received from the AP:

$$E_i^{loc} = k_i f_i^3 t_i \leq \mu P c_i a T \leq \mu P c_i T < k_i f_{\max}^3 T, \ i \in U_0 \tag{5}$$

Therefore, the residual energy $E_i^{res}$ of the $i-$th UAV in a single time block can be expressed as:

$$E_i^{res} = E_i - E_i^{loc} = \mu P c_i a T - k_i f_i^3 t_i, \ i \in U_0 \tag{6}$$

*3.3. Offloading Computing Model*

Let $R_j$ denote the computing task size (bit) of the $j-$th UAV. Since communication overhead is generated during the offloading process, the number of offload task bits received by the AP can be expressed as:

$$R_j^{off} = \theta R_j, \ j \in U_1 \tag{7}$$

where $\theta > 1$ is the communication overhead coefficient, such as the message header or encryption.

According to the Shannon formula of communication engineering knowledge, the offloading transmission power $P_j$ is

$$P_j = \frac{G}{c_j}(2^{\frac{\theta R_j}{b_j T B}} - 1), \ j \in U_1 \tag{8}$$

where $B$ denotes the communication bandwidth, $P_j$ denotes the offload transmission power from the $j-$th UAV to the AP, and $G$ denotes the Gaussian noise power in the channel from the $j-$th UAV to the AP. Therefore, the residual energy of the $j-$th UAV in a single time block is:

$$E_j^{res} = E_j - E_j^{off} = \mu P c_j a T - \frac{G b_j T}{c_j}(2^{\frac{\theta R_j}{b_j T B}} - 1), \ j \in U_1 \tag{9}$$

## 4. Problem Formulation and Solution

*4.1. Problem Formulation*

The research goal of this paper is to maximize the residual energy of the UAV through joint optimization, so it is necessary to maximize the energy acquisition and minimize the energy consumption of the UAV. The energy acquired and consumed by the UAV is related to the choice of its computing mode, wireless communication and system resource allocation. Therefore, for the WPT-MEC with multiple UAVs, the residual energy maximization problem in each time block can be mathematically formalized as problem $P1$:

$$P1: \max_{a, b_j, f_i, t_i, U_0, U_1} \sum_{n \in U} E_n - \sum_{i \in U_0} \sum_{j \in U_1} (E_j^{loc} + E_j^{off})$$

$$s.t. S1: \frac{R_i^{loc} O}{T} \leq f_i \leq \min[f_{\max}, (\frac{\mu P c_i a T}{k_i})^{\frac{1}{3}}], \ i \in U_0$$

$$S2: 0 \leq t_i \leq T, \ i \in U_0$$

$$S3: E_i - E_i^{loc} \geq 0, \ i \in U_0$$

$$S4: a + \sum_{j \in U_1} b_j \leq 1, \ j \in U_1 \tag{10}$$

$$S5: 0 \leq a \leq 1$$

$$S6: 0 \leq b_j \leq 1, \, j \in U_1$$

$$S7: E_j - E_j^{off} \geq 0, \, j \in U_1$$

$$S8: U_0 \subseteq U, U_1 = U \backslash U_0$$

Problem $P1$ describes the optimization objective of this paper in detail and lists the relevant constraints. $S1$ denotes the CPU computing speed constraint in the UAV. $S2$ denotes the computing time constraint in the local computing mode. $S3$ denotes the energy consumption constraint in the local computing mode. $S4$, $S5$ and $S6$ are system energy transmission and offloading time constraints in TDMA mode. $S7$ denotes the UAV energy consumption constraint in offload computing mode. $S8$ denotes a mutual exclusion constraint between the two computing modes.

### 4.2. Problem Analysis and Solution

In problem $P1$, the objective function contains four unknown variables and the selection of combination mode, so the problem is non-convex. However, we can simplify the problem by analyzing the problem $P1$.

In the local computing mode, it can be seen from Formula (3) that $f_i$ is inversely proportional to $t_i$; that is, the larger $t_i$, the smaller $f_i$. It can be seen from Formula (6) that when $f_i$ is smaller, $E_i^{res}$ is larger. Therefore, the conclusion of $t_i = T$ and $f_i = \frac{R_i^{loc}O}{T}$ can be drawn, which can eliminate the $S1$ and $S2$ constraints in the problem $P1$.

Because the goal of this paper is to maximize the residual energy of all UAVs, it is necessary to maximize the energy harvesting time to fully utilize the entire time block. The objective optimal solution should be obtained when the $S4$ equal sign holds—that is, $a^* + \sum_{j \in U_1} b_j{}^* = 1$, where $a^*$ and $b_j^*$ are the optimal time allocation.

The $P1$ problem is an MIP non-convex problem, which is difficult to solve. However, it can be seen through analysis that once $U$ is given, the $P1$ problem can be simplified to a convex problem, which can be solved by convex optimization technology. Therefore, we divide the $P1$ problem into two sub-problems: one is the computing task offloading decision problem, and the other is the time resource allocation problem $P2$:

$$P2: \max_{b_j} \alpha \sum_{n \in U} c_n - [\alpha \sum_{n \in U} \sum_{j \in U_1} c_n b_j + \beta \sum_{i \in U_0} k_i R_i^{loc^3} + \sum_{j \in U_1} \frac{b_j}{c_j} \Psi(\frac{1}{b_j})]$$

$$s.t. \ S3: E_i - E_i^{loc} \geq 0, \, i \in U_0$$

$$S5: 1 - \sum_{j \in U_1} b_j \geq 0, \, j \in U_1 \qquad (11)$$

$$S6: b_j \geq 0, \, j \in U_1$$

$$S7: E_j - E_j^{off} \geq 0, \, j \in U_1$$

where $\alpha = \mu P T$ and $\beta = \frac{O^3}{T^2}$ are fixed parameters, function $\Psi(x) = GT(2^{\frac{\theta R_j x}{B T}} - 1)$.

#### 4.2.1. Computing Task Offloading Decision Problem

There are $N$ UAVs. Thus, there are $2^N$ offloading strategies in total. We need to find the optimal or satisfactory suboptimal offloading strategy. In Section 5, we will solve this problem with the DOTO algorithm.

#### 4.2.2. Time Resource Allocation Problem

Problem $P2$ can be proven to be a convex optimization problem. The fourth term in the objective function $F(x) = \frac{1}{x c_j} \psi(x)$ of problem $P2$ is continuous in the range of

$x \in [0, 1]$. It has first and second derivatives in $x \in (0, 1)$, and $F(x)'' > 0$ can be obtained by derivation. Therefore, the fourth term in the objective function is a concave function, and after adding the minus sign before the parentheses, the fourth term becomes a convex function. The second term is a linear function of variables $b_j$, and the remaining first and third terms do not contain variables, so the objective function of $P2$ is a convex function. At the same time, because the corresponding constraints are convex, problem $P2$ is a convex optimization problem. We can use the Lagrangian method and the KKT condition in the convex optimization method [39] to solve it. Moreover, the Lambert function is used [40]. After solving, it can be obtained as follows:

$$b_j^* = \frac{\theta\, R_j \ln 2}{G\, T(W\left(-\frac{1}{e} + \frac{\mu P\, c_n\, c_j}{e\, G}\right) + 1)} \tag{12}$$

$$a^* = 1 - \sum_{j \in U_1} \frac{\theta\, R_j \ln 2}{GT\left(W\left(-\frac{1}{e} + \frac{\mu P\, c_n c_j}{e\, G}\right) + 1\right)} \tag{13}$$

Thus far, we have solved the optimal time allocation under the given task offloading decision $U$.

## 5. The DOTO Algorithm

### 5.1. Algorithm Overview

In this section, we will design an online computing offloading strategy function $\Omega$ based on deep reinforcement learning in WPT-MEC with a parameter of $\omega$. Input the wireless channel gain $c_t$ at the beginning of each time block, and the online computing offloading strategy function can immediately output the optimal offloading decision. The policy function is expressed as:

$$\Omega_\omega : c \rightarrow u^* \tag{14}$$

We introduce $N$ auxiliary variable $u = [u_1, \ldots, u_n]$, where $u_i = 0$ (or $u_i = 1$) denotes that the computing mode of the $n-$th UAV is local computing, i.e., $i \in U_0$ (or offloading computing, i.e., $i \in U_1$). For each group of offloading actions $u$, the optimal time resource allocation and the corresponding maximum residual energy $E$ can be obtained by solving problem $P2$. After all the generated offloading actions are calculated, the group with the largest residual energy is selected through comparison. The DOTO algorithm will gradually learn the policy function from experience and constantly iterate and update to allow the strategy function $\Omega$ to achieve the best effect. The structure of the DOTO algorithm is shown in Figure 3.

The DOTO algorithm consists of two stages. The first stage is the iteration of the offloading function, and the second stage is the update of the offloading policy. The two stages are alternately performed until the training converges.

In the offload function iteration, we use DNN to generate the offload action, and the DNN contains built-in parameters. Specifically, at the beginning of the $t-$th time block, the wireless channel gain $c_t$ is input, and the DNN outputs a relaxed offloading action $\widetilde{u}_t$ based on the current parameter $\omega_t$ and the offloading strategy $\Omega_{\omega_t}$. Each number in $\widetilde{u}_t$ is between 0 and 1. Then, $\widetilde{u}_t$ is quantized into $K$ binary offload actions. By solving the $P2$ problem, a set of decisions $u_t^*$ with the largest residual UAV energy that can satisfy all constraints among the $K$ offloading decisions is calculated. The optimal time resource allocation is $(a_t^*, b_t^*)$. The MEC system executes the offloading action $u_t^*$, and the DOTO algorithm accepts the reward $E^*(c_t, u_t^*)$, and puts the state-action $(c_t, u_t^*)$ into the experience replay pool. The new offloading strategy generates a new optimal offloading decision $u_{t+1}^*$ based on the new input $c_{t+1}$ in the next time block. With continuous training iterations, the DNN offloading strategy will gradually improve until the training converges. We will describe these two stages in detail below.

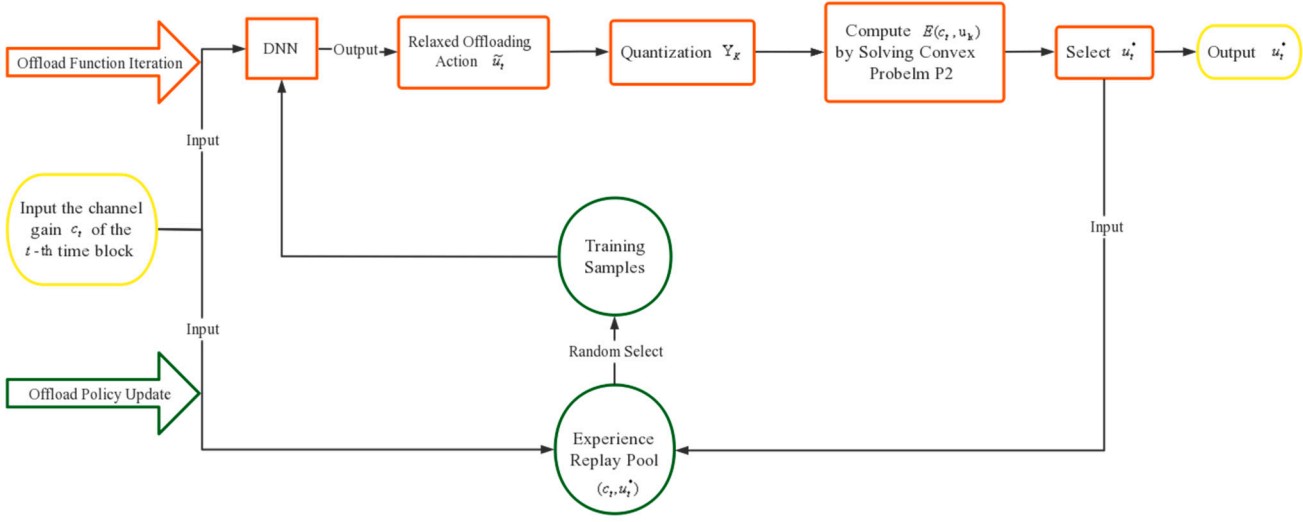

**Figure 3.** The structure of the DOTO algorithm.

*5.2. Offload Function Iteration*

At the $t-$ th time block, the wireless channel gain is $c_t$, where $t = 1, 2, \ldots$. When $t = 1$, the parameter $\omega_t$ of the DNN is randomly initialized according to the standard normal distribution with zero mean. The DNN outputs the first offloading action according to the formula $\widetilde{u}_t = \Gamma_{\omega_t}(c_t)$, which is expressed by the following formula:

$$\widetilde{u}_t = \{\widetilde{u}_{t,i} | \widetilde{u}_{t,i} \in (0,1), i = 1, \ldots, N\} \tag{15}$$

In the hidden layer of the neural network, this paper chooses the ReLU function as the activation function. To relax the offloading action of the output, the Sigmoid function is used as the activation function in the output layer. In this paper, $\widetilde{u}_t$ is quantified into $K$ groups of binary offloading actions by the KNN method, and each group has $N$ binary data. Let the quantization function $\mathrm{Y}_K$ be:

$$\mathrm{Y}_K : \widetilde{u}_t \to \{u_k | u_k \in \{0,1\}_N, k = 1, \ldots, K\} \tag{16}$$

There are a total of $N$ UAVs, so the value range of $K$ is $K \in [1, 2^N]$. When $K$ is larger, the number of calculations for the residual energy problem will be greater, and the computational complexity of the DOTO algorithm will be higher, but the quality of the decision-making scheme will be better. Similarly, the smaller the $K$, the lower the computational complexity and the worse the quality of decision-making scheme. In general, setting a larger $K$ can obtain more UAV residual energy at the cost of higher complexity. However, if a large number of quantization actions are generated in each time block, it is very unnecessary and the efficiency will be very low. Therefore, in order to balance the quality of the algorithm decision-making scheme and the computational complexity, we propose an adaptive $K$ value setting method. The details are as follows:

(1)    Initially, set the $K$ value to $2^N$. During training, all alternatives generated by the KNN quantization method are sorted by Euclidean distance.

(2)    Each time block records the index value $K$ corresponding to the maximum residual energy offloading action. Every $\Delta$ time block, compare the largest $K_\Delta$ in the $\Delta$ time block with the $K_{\Delta-1}$ in the $\Delta - 1$ time block, and select the largest $K + 1$ value as the next iteration.

(3)    In order to avoid excessive training loss caused by an overly small $K$ value, when $K < N$, we set the $K$ value as $N$, i.e., the value range of $K$ is $K \in [N, 2^N]$.

The adaptive *K* value setting is expressed by the formula below:

$$K = \begin{cases} 2^N, t = 1 \\ \max(\max(k_{t-1}, \ldots, k_{t-\Delta}) + 1, N), t \bmod \Delta = 0 \end{cases} \tag{17}$$

After obtaining *K* offloading decisions at the $t-$th time block, we can calculate the UAV residual energy corresponding to each offloading decision by solving the *P*2 problem. Therefore, the optimal offloading decision $u_t^*$ corresponding to the maximum residual energy is selected, which can be expressed as follows:

$$u_t^* = \arg \max_{u_i \in \{u_K\}} E^*(c_t, u_i) \tag{18}$$

Finally, output the optimal offloading decision.

*5.3. Offload Policy Update*

In order to reduce the correlation between data and improve the learning efficiency, we introduce the experience replay technology, which can effectively improve the generalization ability of the algorithm and accelerate the convergence speed of the algorithm. Firstly, we create an empty memory with limited capacity, and add the wireless channel gain $c_t$ and optimal offloading decision $u_t^*$ to the experience replay pool in the $t-$th time block. When the memory capacity of the experience replay pool is full, the old data samples will be updated with the newly generated data samples. Therefore, the DNN will only learn relatively new and better offloading decision samples.

At the next time block, we randomly select a batch of training data samples $\left\{ (h_\rho, u_\rho^*) \,|\rho \in I_t \right\}$ from the experience replay pool to train the DNN, where $I_t$ denotes the total number of time blocks in the experience playback pool. In the DNN, we use the Sigmoid activation function, so it is suitable to use the cross-entropy loss function at this time. The main characteristic of the cross-entropy loss function is that when the error is large, the weight update is fast, and when the error is small, the weight update is relatively slow. The formula is as follows:

$$Loss(\omega_t) = -\frac{1}{|I_t|} \sum_{\rho \in I_t} \left\{ (u_\rho^*)^{\mathrm{T}} \log \Gamma_{\omega_t}(c_t) + (1 - u_\rho^*)^{\mathrm{T}} \log[1 - \Gamma_{\omega_t}(c_t)] \right\} \tag{19}$$

where the parameter $\omega_t$ of the DNN reduces the cross-entropy loss by using the Nadam optimization algorithm [41,42]. After bias correction, the Nadam algorithm will set different adaptive learning rates for different parameters, and the learning rate will have a certain range, so that the parameters are relatively stable. Based on the advantages of the RMSprop algorithm for dealing with non-stationary targets, Nadam also has the characteristics of the Adagrad algorithm for dealing with sparse gradients. It is equivalent to Adam with Nesterov momentum and has great advantages in solving problems with large-scale data or parameters.

In general, the DNN continuously learns from optimal state actions $(c_t, u_t^*)$, producing increasingly superior offloading decision outputs over time. It continuously improves the offloading strategy under the mechanism of reinforcement learning until it converges, thus forming an online computing offloading strategy suitable for the MEC system. The DOTO algorithm is shown in Algorithm 1.

---

**Algorithm 1** The DOTO algorithm

---

**Input:** $c_t$: wireless channel gain at each time block $t$; $M$: the number of time
blocks; $\delta$: training interval; $K$: the number of quantized actions;
**Output:** optimal offloading decision $u_t^*$;
1: Initialize the DNN with random parameters $\omega_1$ and empty memory;
2: **for** $t = 1, \ldots, M$ **do**
3: Generate a relaxed offloading action $\widetilde{u}_t = \Gamma_{\omega_t}(c_t)$;
4: Quantize $\widetilde{u}_t$ into $K$ binary actions $\{u_k\} = Y_K(\widetilde{u}_t)$;
5: Compute $E^*(c_t, u_k)$ for all $u_k$ by solving $P2$;
6: Select the optimal offloading decision $u_t^* = \arg \max\limits_{u_i \in \{u_K\}} E^*(c_t, u_i)$;
7: Update experience replay pool by adding $(c_t, u_t^*)$;
8: **if** $t \bmod \delta = 0$ **then**
9: Randomly select training samples $\left\{ (h_\rho, u_\rho^*) \, | \rho \in I_t \right\}$;
10: Update $\omega_t$ using the Nadam algorithm;
11: **end**
12: **if** $t \bmod \Delta = 0$ **then**
13: Update $K$ by $K = \max(\max(k_{t-1}, \ldots, k_{t-\Delta}) + 1, N)$;
14: **end**
15: **end**

---

## 6. Simulation Results

In this section, we will use simulation experiments to evaluate the performance of the DOTO algorithm. In order to be in line with reality, the parameters are set as follows in this paper [8,20]. In all simulation experiments, the transmitter parameter of the AP server is set to $P = 50W$, the UAV energy harvesting efficiency $\eta_i$ is uniformly distributed in the $[0.6, 0.8]$ interval, and the distance $d_i$ from each UAV to the AP is uniformly distributed in the range of $(5,8)$ meters, $i = 1, \ldots, N$. The channel gain $\widetilde{c}_i$ follows the free-space path loss model $\widetilde{c}_i = A_d \left( \frac{3 \times 10^8}{4 \pi h d_i} \right)^{d_e}$, where $A_d = 4.11$ denotes the antenna gain, $d_e$ denotes the path loss index, generally $d_e = 2.8$, and $h = 915 \, MHz$ represents the carrier frequency. In the $t - $th time block, the wireless channel gain $c_t = [c_t^1, c_t^2, \ldots, c_t^N]$ of $N$ UAVs can be obtained from the Rayleigh fading channel model $c_t^i = \widetilde{c}_i \alpha_i^t$, where $\alpha_i^t$ represents an independent random channel fading factor and obeys the exponential distribution of unit mean. We use the TDMA protocol for computing task offloading. To ensure the generality of the experiment, the length of the time block is set to be less than the time of the channel interference—that is, $T = 1$. It is assumed that the wireless channel gain remains constant in the same time block, but changes in different time blocks. We assume that the computing efficiency $k_i$ of all UAVs is equal, which is $k_i = 10^{-22}$, $i = 1, \ldots, N$, and the period $O = 200$ cycles/bit. The data communication bandwidth is $B = 2 \, MHz$, the noise power is $G = 10^{-10}$, and the offload task communication overhead coefficient $\theta = 1.1$. The DNN model consists of two hidden layers, an input layer and an output layer, and the training environment is Tensorflow2. Below, we analyze the experimental results.

### 6.1. Performance of Algorithm

For the problem of the UAV offloading strategy in the MEC system presented in this paper, the Coordinate Descent (CD) method randomly gives a set of offloading actions for all UAVs. For the combination of each piece of equipment in offloading mode and local calculation mode in sequence, the maximum residual energy of the UAV is calculated by solving formula $P2$. After comparison, the computing mode with the largest residual energy in the UAV is selected. When the mode conversion is performed for a device, the offloading action of other devices is fixed. The CD algorithm is equivalent to searching in a two-dimensional space at this time. The dimension is low and the result is good.

In this section, we evaluate the decision quality of the DOTO algorithm. Firstly, the optimal offloading decision of each time block is found from all $2^N$ offloading actions of $N$ UAVs. Then, we find the total residual energy of the UAV in the MEC system obtained

within 10,000 time blocks under the three modes of the DOTO algorithm, CD algorithm and optimal offloading decision. In Figure 4, we compare the ratio of the results of the DOTO algorithm and the CD algorithm to the results of the optimal offloading decision. The closer the curve in the figure is to 1, the better the algorithm's performance. From the results in Figure 4, it can be seen that when the number of UAVs in the MEC system varies between 5 and 10, the DOTO algorithm performs better than the CD algorithm. When the number of devices increases, the performance of the two modes decreases gradually. However, the trend of performance degradation for the DOTO algorithm is more gradual than that of the CD algorithm. When the number of devices increases, the performance gap between the two algorithms is larger. When there are 10 UAVs, the DOTO algorithm can still obtain 99.9% of the residual energy. However, the CD algorithm can obtain 99.2% of the residual energy because, when the number of devices increases, the possible offloading actions increase exponentially. At this time, the simple two-dimensional search of the CD algorithm is unable to search for the global optimal solution, so the performance of the algorithm becomes increasingly worse. The DOTO algorithm benefits from the powerful fitting effect of the DNN, and even if the number of devices is large, the data sample training is sufficient. Therefore, the performance of the algorithm is relatively better and more stable.

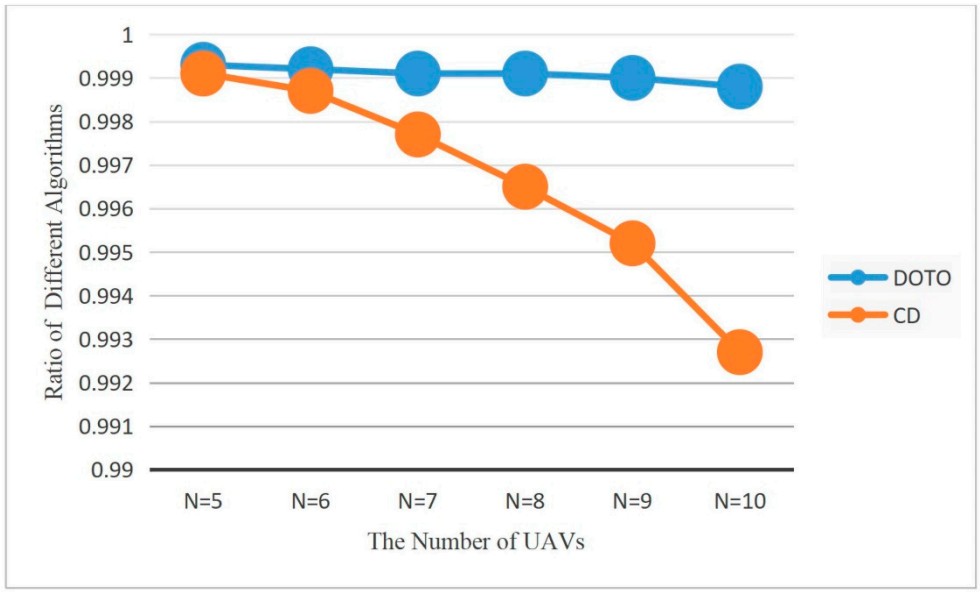

**Figure 4.** The ratio of the results of the DOTO algorithm and the CD algorithm to the results of the optimal offloading decision.

### 6.2. Adaptive K Value Setting Method

In this section, we study the impact of the adaptive *K* value setting method on the performance of the DOTO algorithm.

In Figure 5, we show the *K* value index distribution in 10,000 time blocks when the number of UAVs accessed in the MEC system is 5 to 10. At this time, the *K* value is mostly 0 or 1, but there are also some other values. When the number of UAVs increases, the final value fluctuates more. However, more than 95% are within the *N* value and below. Therefore, in the adaptive *K* value setting method, this paper finally sets the *K* minimum to the *N* value. The experiments in Section 6.1 have demonstrated that the DOTO algorithm with such an adaptive *K* value setting can obtain satisfactory suboptimal solutions.

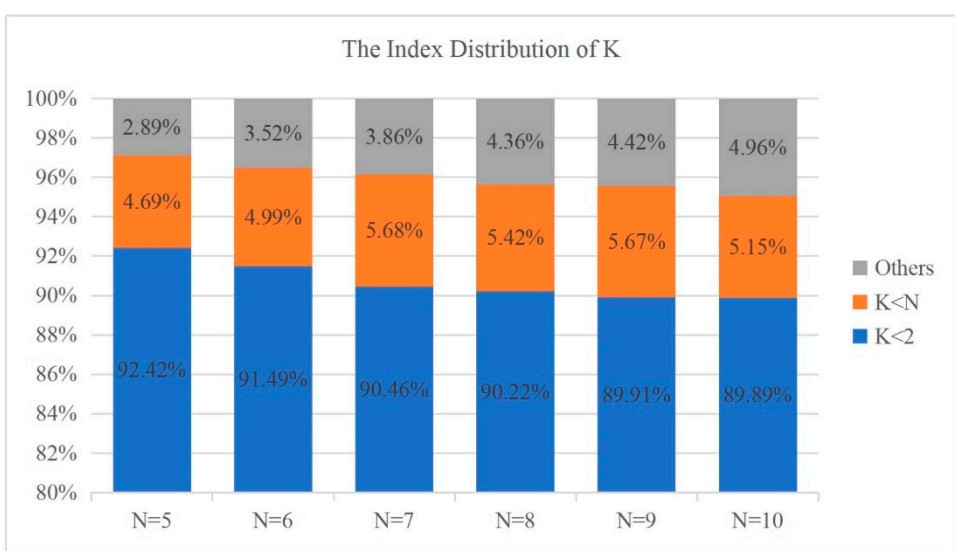

**Figure 5.** The index distribution of the *K* value of the maximum offloading action in KNN within 10,000 time blocks when the number of UAVs varies from 5 to 10.

In Figure 6, we show the influence of the adaptive *K* on the computational complexity of the DOTO algorithm. It shows that the computational complexity of the DOTO algorithm can be greatly reduced under the condition of ensuring the performance of the algorithm. When the value of *K* is $2^N$, as the number of UAVs in the MEC system increases, the time consumed by the offloading strategy will increase significantly. When the number of access devices is 10, the average time required for each offloading decision reaches 0.1203 s. At this time, the delay is too high, and it is no longer suitable for solving practical problems. Moreover, when the number of devices continues to increase, the delay will be more serious. When the adaptive *K* value setting is adopted, the average calculation time of the DOTO algorithm is relatively stable, and there is no obvious change. The required time is also approximately 0.017 s, which is short and can be well applied to the actual model. The computational complexity of the DOTO algorithm is reduced from an exponential relationship to a linear relationship.

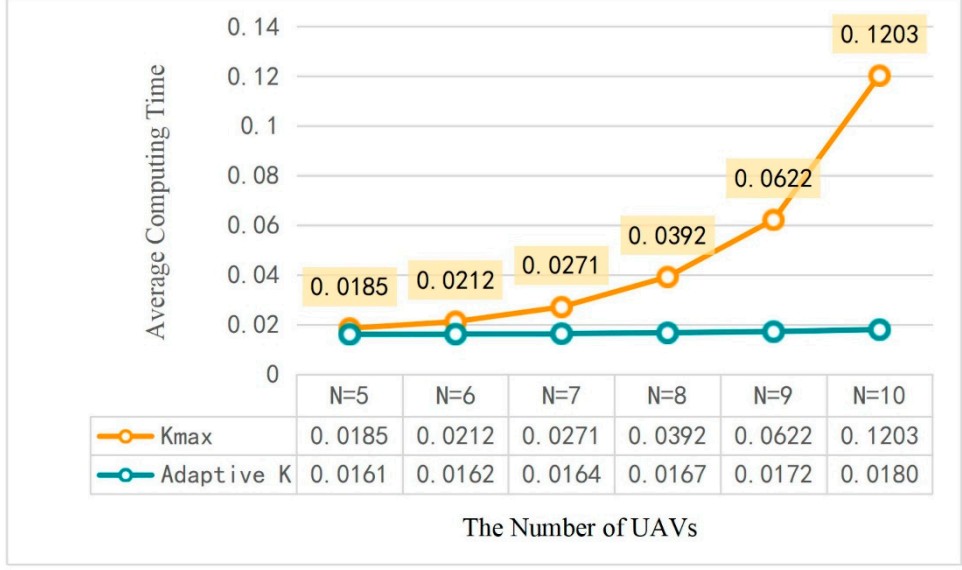

**Figure 6.** When the number of UAVs is 5 to 10, the DOTO algorithm uses full enumeration ($K_{max}$) and adaptive *K* modes to average the computing time per time block within 10,000 time blocks.

In Figure 7, we further demonstrate the effect of the adaptive $K$ update interval $\Delta$ on the computational complexity and residual energy of the DOTO algorithm. To make the image more intuitive, we divide the data obtained at different update intervals $\Delta$ by the smallest data among them. This converts the data into ratio form. The minimum computing time and residual energy in the figure are obtained when the update interval $\Delta$ is 5. Therefore, the data in the figure are all obtained by dividing the data when the update interval is 5. It can be seen from Figure 7 that when the update interval $\Delta$ is larger, the required calculation time is greater and the calculation complexity is higher. However, the residual energy obtained is also greater because, when the update interval $\Delta$ is larger, the $K$ value decreases more slowly. This means that more offloading actions will be generated and problem *P*2 will be calculated more times. However, when the update interval $\Delta \geq 20$, the computation time grows exponentially, while the residual energy grows linearly and slowly. To balance performance and computational complexity, we finally set $\Delta = 40$.

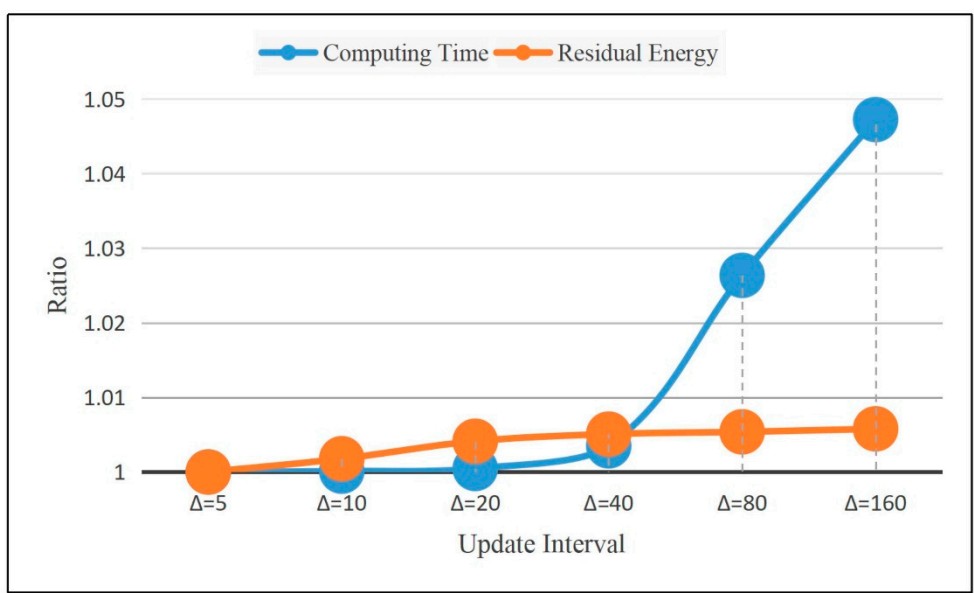

**Figure 7.** The corresponding calculation time and residual energy when adaptive $K$ adopts different update intervals $\Delta$. Here, we set $N = 8$.

### 6.3. The Effects of Different Parameters

In this section, we study the effects of different parameters on the performance of the DOTO algorithm, including the training interval, experience replay pool and training sample.

In Table 1, we show the performance of DOTO under different training intervals $\delta$. The DOTO algorithm converges faster with shorter training intervals, and thus more frequent policy updates. However, it is not necessary to update the policy frequently. The results in Table 1 also prove this point. Nonetheless, the training interval also should not be too large. This will cause the algorithm not to converge. Since the DOTO algorithm no longer converges when the training interval $\delta \geq 40$, the data results are meaningless and are thus not displayed. After comprehensively considering the algorithm results and convergence performance, this paper sets the training interval $\delta = 10$.

In Tables 2 and 3, we show the performance of DOTO under different experience replay pool sizes and training sample sizes. First, when the training sample size is 64, 128 and 256, we use different experience replay pool sizes to observe the performance of the algorithm. Then, after selecting a suitable size of the experience replay pool, we find a suitable training sample size according to the size of the experience replay pool. It can be seen from Table 2 that when the experience pool is small, the convergence of the DOTO algorithm fluctuates greatly. When the experience pool is larger, the convergence is slower. After comprehensively considering the algorithm results and convergence performance, we finally set the experience replay pool size to 1024.

**Table 1.** The effect of training interval $\delta$ on algorithm performance. Here, we set $N = 8$.

| Training Interval | Ratio of Residual Energy to Optimal Decision | Convergence Performance |
|---|---|---|
| 2 | 0.9985 | convergence fluctuates greatly |
| 5 | 0.9990 | convergence fluctuates greatly |
| 10 | 0.9991 | convergence |
| 20 | 0.9990 | convergence |
| 40 | | non-convergence |
| 80 | | non-convergence |

**Table 2.** The effect of experience replay pool size on algorithm performance. Here, we set $N = 8$.

| Experience Replay Pool | Training Sample | Ratio of Residual Energy to Optimal Decision | Convergence Performance |
|---|---|---|---|
| 64 | 64 | 0.9987 | convergence fluctuates greatly |
| 128 | 64 | 0.9986 | convergence fluctuates greatly |
| | 128 | 0.9984 | convergence fluctuates greatly |
| 256 | 64 | 0.9988 | convergence |
| | 128 | 0.9988 | convergence fluctuates greatly |
| | 256 | 0.9987 | convergence fluctuates greatly |
| 512 | 64 | 0.9988 | convergence |
| | 128 | 0.9989 | convergence fluctuates greatly |
| | 256 | 0.9989 | convergence fluctuates greatly |
| 1024 | 64 | 0.9990 | convergence |
| | 128 | 0.9991 | convergence |
| | 256 | 0.9990 | convergence |
| 2048 | 64 | 0.9989 | slow convergence |
| | 128 | 0.9991 | slow convergence |
| | 256 | 0.9990 | convergence |

As shown in Table 3, when the training samples are too small, since DOTO cannot effectively utilize all the data in the experience pool, the convergence performance of the algorithm is reduced. When the number of samples is too large, old training data are often used, resulting in poor convergence performance. After comprehensively considering the algorithm results and convergence performance, we finally set the training sample size to 128.

**Table 3.** The effect of training sample size on algorithm performance. Here, we set the experience replay pool size to 1024 and $N = 8$.

| Training Sample | Ratio of Residual Energy to Optimal Decision | Convergence Performance |
|---|---|---|
| 16 | | non-convergence |
| 32 | 0.9985 | convergence fluctuates greatly |
| 64 | 0.9990 | convergence |
| 128 | 0.9991 | convergence |
| 256 | 0.9990 | convergence |
| 512 | 0.9988 | convergence fluctuates greatly |
| 1024 | 0.9985 | convergence fluctuates greatly |

*6.4. Switching UAV On or Off*

During the operation of the MEC system, individual UAVs may shut down due to malfunction or loss of power. New UAVs may also be connected to the system. In this section, the stability of the DOTO algorithm will be analyzed when individual devices are turned off or new devices are connected during operation.

In Figure 8, we show whether the training loss converges quickly after turning the device on or off. When the number of iterations is 400, one UAV is randomly turned off in the MEC system with six UAVs, and two UAVs are newly connected in the system with seven UAVs. At this time, the training loss fluctuates greatly, rising (or falling) rapidly, but it converges again after around 100 iterations. This shows that the DOTO algorithm automatically updates its own offloading strategy and converges to obtain the optimal offloading strategy under the new environmental state.

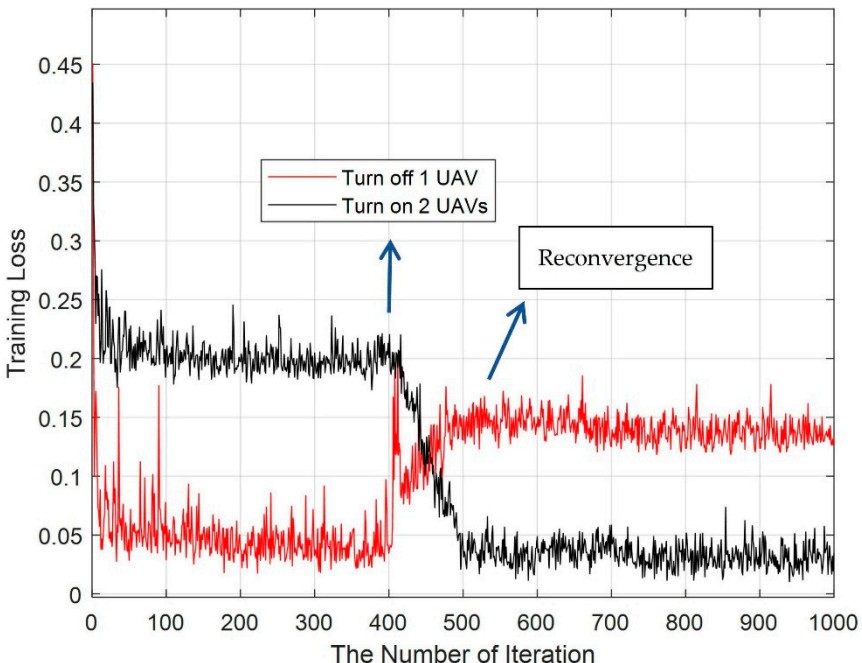

**Figure 8.** Training loss obtained by turning UAV on or off in the MEC system at iteration 400.

In Figure 9, we further show the more complex case where one of the UAVs is randomly turned off and then turned on in an MEC system with 10 UAVs. Specifically, when the number of training steps is 900, one UAV is randomly turned off. Moreover, when the number of training steps is 1800, it is turned on and re-connected to the system. It can be

seen from Figure 9 that the training is in a convergent state at first. However, after randomly shutting down one device, the training loss fluctuates greatly and rises rapidly, reaching convergence again after around 100 steps of training. When the number of training times is less than 1800, we reopen the original device to access the MEC system. At this point, the training loss fluctuates again and quickly converges to the state of the original 10 devices. This shows that the DOTO algorithm has strong self-adjustment ability and can be well suited for complex MEC online offloading situations.

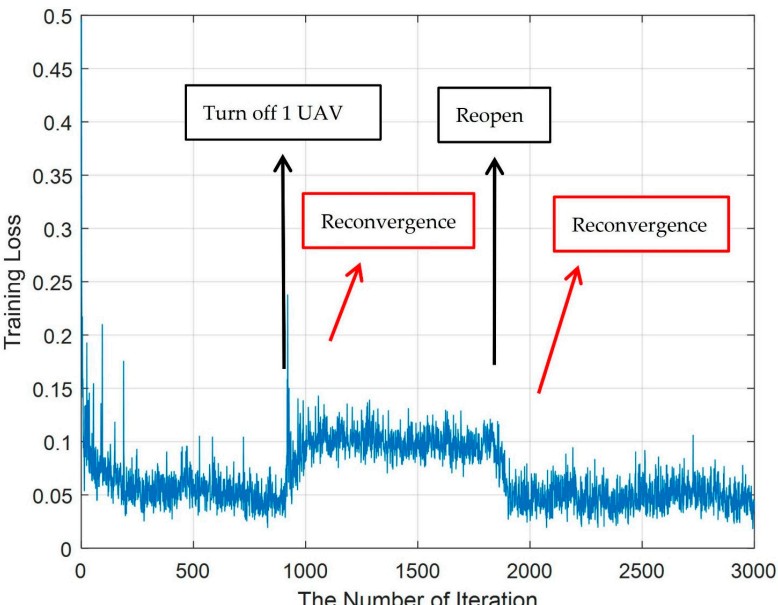

**Figure 9.** The convergence performance of the training loss when randomly turning one UAV off and then reopening it.

The above two experiments fully demonstrate that the DOTO algorithm has good adaptability.

## 7. Conclusions

This paper studies the problem of maximizing the residual energy of maritime search and rescue UAVs based on MEC. The research objective is formalized as a residual energy maximization problem by jointly optimizing communication and time resources. After analysis, the problem is found to be a non-convex problem, which is difficult to solve. However, when the offloading decision is given, it can be transformed into a convex problem and solved by convex optimization techniques. Therefore, we divide the target problem into two sub-problems, namely the time resource allocation problem and the task offloading decision problem. For the time resource allocation problem, the relevant optimal parameters can be obtained by using the convex optimization method. For the task offloading problem, the DOTO algorithm is proposed based on the idea of deep reinforcement learning. To balance the performance and computational complexity of the algorithm, an adaptive $K$ value method is proposed. Simulation experiments show that the DOTO algorithm can provide users with an online computing offloading strategy that is superior to other traditional benchmark schemes. When there is a UAV in the MEC system that exits the system due to insufficient power or failure, or a new UAV is connected to the system, the DOTO algorithm can also quickly converge to provide a new and effective online offloading strategy for the MEC system. There is no need for manual participation in the whole process, resulting in good stability. Deep reinforcement learning provides a new solution for MEC. In the future, we will explore the application of deep reinforcement learning for the joint offloading of multiple APs. We will also focus on the balance between the residual energy and computing rate of mobile devices.

**Author Contributions:** Conceptualization, Z.D. and J.G.; methodology, Z.D. and G.X.; software, Z.D. and Z.L.; validation, Z.D. and Z.L.; investigation, Z.D. and Z.L.; resources, J.G. and W.W.; data curation, Z.D.; writing—original draft preparation, Z.D.; writing—review and editing, Z.D., J.G., Z.L. and W.W.; visualization, Z.D.; supervision, G.X.; project administration, G.X. All authors have read and agreed to the published version of the manuscript.

**Funding:** This work was supported by the Development Project of Jilin Province of China (No. 20200401076GX) and Jilin University of China.

**Data Availability Statement:** Not applicable.

**Conflicts of Interest:** The authors declare no conflict of interest.

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
