# Peer review of "Energy Saving Strategy of UAV in MEC Based on Deep Reinforcement Learning"

_futureinternet, doi:10.3390/fi14080226_

Round 1

Reviewer 1 Report

This paper combines UAV and mobile edge computing (MEC), and designs an online task offloading algorithm based on deep reinforcement learning (DOTO).  The proposed algorithm can obtain an offloading strategy that maximizes the residual energy of UAV by jointly optimizing UAV time and communication resources.  

Paper's strengths:  1-The contributions are well-presented, and the background is clearly explained 2-The paper is well organized and easy to follow  

paper's weaknesses: 1- the presented symbols in the equations are confusing. 2- more details regarding the proposed method implementation and evaluation are needed  

Comments: -define any acronyms you use for the first time in the main text (even if you have already defined them in the abstract). e.g. MEC, UAV  ...etc. -in equation (1), please add space between Mu P and Cn, and other symbols, now it reads as MuPCn as a single symbol. The same goes for (6) and other equations. -in Section 6, you have defined the parameters, however, you did not give any explanation/justification why these values are preferable? -the following related works can be added:

[1] Zhao, Nan, et al. "Multi-Agent Deep Reinforcement Learning for Task Offloading in UAV-assisted Mobile Edge Computing." IEEE Transactions on Wireless Communications (2022).

[2]Naouri, Abdenacer, Hangxing Wu, Nabil Abdelkader Nouri, Sahraoui Dhelim, and Huansheng Ning. "A novel framework for mobile-edge computing by optimizing task offloading." IEEE Internet of Things Journal 8, no. 16 (2021): 13065-13076.

[3] Abd Elaziz, Mohamed, Laith Abualigah, Rehab Ali Ibrahim, and Ibrahim Attiya. "IoT workflow scheduling using intelligent arithmetic optimization algorithm in fog computing." Computational intelligence and neuroscience 2021 (2021).

[4] Zhang, Lu, et al. "Task offloading and trajectory control for UAV-assisted mobile edge computing using deep reinforcement learning." IEEE Access 9 (2021): 53708-53719.

[5] Dhelim, Sahraoui, Huansheng Ning, Fadi Farha, Liming Chen, Luigi Atzori, and Mahmoud          

Author Response

1. Spaces have been added to equations, as in equation (1), (6) and others.

2. Acronyms used for the first time in the main text are defined. e.g. MEC, UAV  ...etc.

3. Regarding the Nadam optimizer in the DOTO algorithm, a new reference has been introduced for clarification.(Line 411)

4. According to the actual scene and simulation experiment results, referring to the relevant literature (such as [8] [13] [16-17] [20]), the parameters of this paper are set.  (Line 48, 53, 143, 428)

5.New references are cited:

[33] Zhao, N.; Ye, Z.; Pei, Y.; Liang, Y.-C.; Niyato, D. Multi-Agent Deep Reinforcement Learning for Task Offloading in UAV-Assisted Mobile Edge Computing. IEEE Transactions on Wireless Communications 2022.

[25] Naouri, A.; Wu, H.; Nouri, N.A.; Dhelim, S.; Ning, H. A Novel Framework for Mobile-Edge Computing by Optimizing Task Offloading. IEEE Internet of Things Journal 2021, 8, 13065–13076.

[4] Abd Elaziz, M.; Abualigah, L.; Ibrahim, R.A.; Attiya, I. IoT Workflow Scheduling Using Intelligent Arithmetic Optimization Algorithm in Fog Computing. Computational intelligence and neuroscience 2021, 2021.

[34] Zhang, L.; Zhang, Z.-Y.; Min, L.; Tang, C.; Zhang, H.-Y.; Wang, Y.-H.; Cai, P. Task Offloading and Trajectory Control for UAV-Assisted Mobile Edge Computing Using Deep Reinforcement Learning. IEEE Access 2021, 9, 53708–53719.

[5] Dhelim, S.; Ning, H.; Farha, F.; Chen, L.; Atzori, L.; Daneshmand, M. IoT-Enabled Social Relationships Meet Artificial Social Intelligence. IEEE Internet of Things Journal 2021, 8, 17817–17828.

Reviewer 2 Report

This paper discusses an online task offloading algorithm based on deep reinforcement learning (DOTO). The algorithm can obtain an online offloading strategy that maximizes the residual energy of UAV by jointly optimizing UAV time and communication resources.

- The topic discussed in this manuscript is interesting and fits the journal's scope. The structure is sound and clear and the evaluation seems to support the authors' claims.

However:

- I would suggest that the authors avoid the use of "etc." in the manuscript. Try to list fully or describe the concept list instead in each case.

- "DOTO" abbreviation in the abstract should be explained.

- In the Related Work section, the authors could include also some other recent works on the field which make deal with TDMA UAV computational offloading for computing tasks at the Edge:

[1] Avgeris, M., Spatharakis, D., Dechouniotis, D., Kalatzis, N., Roussaki, I., & Papavassiliou, S. (2019). Where there is fire there is smoke: A scalable edge computing framework for early fire detection. Sensors, 19(3), 639.

- Short forms like "can’t" should generally be avoided in an academic manuscript as they are too informal.

- Make sure you explain every abbreviation used (e.g., "RF" in line 49).

- Is the technology of wireless power transmission from an AP to a UAV a real thing? are there any real-world implementations or applications of this technology?

- In section 5.1, the authors claim that "The two stages are alternately performed until the training converges". Are there any guarantees that this alternating between the two stages actually converges?

- Why is a Deep Learning method required here? isn't a simple Learning method enough to solve this problem? A short discussion on the number of possible decisions as the algorithm's outcome should be made, to justify the Deep Learning aspect. 

- Line 392: Nadam Algorithm needs a reference from the literature. 

- Line 397: Adam and Nesterov need a reference as well.

Author Response

  1. ‘’etc.‘’ and "can't" are no longer used in the text.
  2. Changed the expression of DOTO to the clearer '’deep reinforcement learning-based online task offloading '',where "D" represents ‘’deep reinforcement learning”, ''O, T, O"represent ‘’online, task, offloading‘’ respectively. (Line 12)
  3. In the related work section, the literature [31] on the combination of TDMA and UAV has been added. (Line 137)

         And new references introduced:

         [6] Avgeris, M.; Spatharakis, D.; Dechouniotis, D.; Kalatzis, N.; Roussaki, I.;               Papavassiliou, S. Where There Is Fire There Is Smoke: A Scalable Edge                   Computing Framework for Early Fire Detection. Sensors 2019, 19, 639.

      4. "RF" has been explained. (Line 51)

      5. There is a lot of work related to wireless power transfer from AP to user                devices, such as references [8] [13] [16-17] [20]. (Line 48, 53,                              143,428)

      6. The stage of "offload function iteration" is the deep learning process. The            stage of "offload policy update" is the reinforcement learning process.                  According to the actual scene and simulation experiment results, the                    parameters are adjusted to make the DOTO algorithm converge.                       ï¼ˆLine 339-341)

       7. Because the traditional method is aimed at offline offloading strategy, it               is not suitable for the environment of fast fading of wireless channel, and             cannot make timely and correct offloading decision. Deep reinforcement             learning is suitable for variable state space and can perceive                                 environmental changes to obtain online offloading strategies.                            (Line 64-68,Line 130-131)

       8. The number of possible decisions in the algorithm is 2N. In this paper, we             propose an adaptive K value setting method, which only selects K                         possible decision, thus balancing the quality of the algorithm decision-                 making scheme and the computational complexity.                                               (Line 300-302, 367-386)

       9. “Adam” and ”Nadam” are the optimization algorithms in the official                     python documentation. So I didn't list its update process. It is explained               by introducing a new reference. (Line 411)

Reviewer 3 Report

This paper studied the problem of maximizing the residual energy of maritime search and rescue UAVs based on MEC. The research objective is formalized as a residual energy maximization problem by jointly optimizing communication and time resources. The paper is interesting supported by the results. However, the following are the concern of the reviewers:

1. There are lots of work focused on Deep Reinforcement Learning for task offloading of UAVs. I think authors should do a literature survey and enlist them in a table and do a comparison that would enhance the main contributions of this work.

2. The system model part lacks details on its use in a real-life scenario. Use some examples and make the system model practically deployable.

3. Where and how Access points should be placed in the maritime environment? What distances should be there for the optimal WPT?

Author Response

  1. There are lots of work focused on deep reinforcement learning for task offloading of UAVs. I have newly added relevant literature [35] and literature [36] and explained its work. (Line 148-155)                                                             
  2. There are many works related to task offloading in the WPT-MEC system, such as references [8] [13] [16-17] [20]. Compared with land, the interference of wireless communication at sea will be less. So we have reason to think that the system model in this paper can be deployed. (Line 48, 53, 143, 428)                   
  3. Access points will be deployed on board and will be supported by the ship. According to the actual application scenario, WPT transmission effect and related references (such as [8] [13-17] [20]), we set the distance to 5 to 8 meters. (Line 48-53, 143,428)

Round 2

Reviewer 1 Report

my comments have been addressed

Reviewer 2 Report

The authors have successfully addressed all my comments. 

Reviewer 3 Report

Accept in present form.